# Organisational Justice and Political Agency among Nurses in Public Healthcare Organisations: A Qualitative Study Protocol

**DOI:** 10.3390/ijerph18179110

**Published:** 2021-08-29

**Authors:** Camelia López-Deflory, Amélie Perron, Margalida Miró-Bonet

**Affiliations:** 1Department of Nursing and Physiotherapy, University of the Balearic Islands, 07122 Palma, Spain; mmiro@uib.es; 2Care, Chronicity, and Health Evidences Research Group, University of the Balearic Islands, 07122 Palma, Spain; 3School of Nursing, Faculty of Health Sciences, University of Ottawa, Ottawa, ON K1N 6N5, Canada; amelie.perron@uottawa.ca; 4Care, Chronicity, and Health Evidences Research Group, Health Research Institute of the Balearic Islands (IdISBa), 07010 Palma, Spain

**Keywords:** nursing, political agency, organisational justice, critical discourse analysis

## Abstract

Nurses are rarely treated as equals in the social, professional, clinical, and administrative life of healthcare organisations. The primary objective of this study is to explore nurses’ perceptions of organisational justice in public healthcare institutions in Majorca, Balearic Islands, Spain, and to analyse the ways in which they exercise their political agency to challenge the institutional order when it fails to reflect their professional ethos. An ethnomethodological approach using critical discourse analysis will be employed. The main participants will be nurses occupying different roles in healthcare organisations, who will be considered central respondents, and physicians and managers, who will be considered peripheral respondents. Data generation techniques include semi-structured interviews, a sociodemographic questionnaire, and the researcher’s field diary. This is one of the first studies to address organisational justice in healthcare organisations from a macrostructural perspective and to explore nurses’ political agency. The results of this study have the potential to advance knowledge and to ensure that healthcare organisations are fairer for nurses, and, by extension, for the patients in their care.

## 1. Introduction

Organisational justice is defined as employees’ perceptions of their organisation’s behaviour, decisions and actions and the influence of these on employees’ own attitudes and behaviour in the workplace [1]. The concept encompasses four dimensions: distributive justice, procedural justice, informational justice, and interactional justice [2]. These dimensions relate to the set of rules and social norms within an organisation that determine how resources should be allocated, which procedures should be used to make decisions, how much information about the decision-making process should be shared, and how employees should be treated [3].

Organisational justice has rarely been studied in the context of healthcare organisations. Of the rare studies carried out in this context, most have opted for a quantitative methodology and have studied nurses, aiming to find significant relationships between the different dimensions of organisational justice and variables such as trust in the organisation [4,5], organisational identification [4] and commitment [6,7,8], job involvement [9], job satisfaction [4,7,10,11], perceived self-efficacy [4], empowerment [12,13], intraprofessional collaboration [14], deviant behaviours [15], organisational citizenship behaviour [16,17,18], violent assaults by patients [14], mobbing [19], moral distress [10,20,21], presence of depressive symptoms [21], health complaints [22] and early retirement intentions [9], among others.

The studies reviewed share two main characteristics. Firstly, they are based on a pre-established definition of organisational justice. With a few notable exceptions [7,12,23,24,25,26], none of the studies allowed participants to define justice and injustice themselves or to describe how they experience these concepts in the organisations in which they work. Secondly, the studies approach healthcare organisations in isolation and fail to consider the links between them and the economic, political, social, and cultural context that influences their functioning. These have been identified as the two main weaknesses in theories of organisational justice and reveal the need to place individuals at the centre of future research in this area by exploring their own perceptions of organisational justice and to include the macrostructural context in order to explain what is happening within the microstructural space of healthcare institutions [27].

As noted, nurses are the most widely studied professional group in research on organisational justice in healthcare organisations. Nurses have traditionally been a subordinate group, experiencing a lack of professional autonomy and power and suffering economic, political, historical, social, and cultural oppression, and continue to be viewed in this way today [28,29]. The empirical reasons for this have been amply explored in the literature and include working conditions characterised by a lack of sufficient material and human resources enabling nurses to perform their professional duties to their expected standard [30,31,32], low social status deriving from the lower value attached to care work (associated with women) [32], failure to fully involve nurses in decision-making in healthcare teams, and poorer access to participation in decision-making on public health [33].

The differences between the working conditions of nurses and other professionals like physicians suggest that nurses are not viewed as equals in healthcare organisations [34], raising questions around equity and justice in these institutions. These aspects are absent from the studies on organisational justice that were reviewed. It is important that they are taken into consideration, however, as inadequate working conditions preventing nurses from carrying out their duties to the best of their abilities represent a form of injustice towards them and the patients in their care [35].

Although the scientific literature has explored the reasons for nurses’ subordination in healthcare organisations, few studies have sought to understand the action taken by nurses to overcome this subordination and make their workplaces fairer for themselves and their patients [36,37,38]. This is linked to the concept of political agency.

Political agency is defined as individuals’ ability to bring about changes to the sociocultural order according to their understanding of their role within the sociopolitical matrix, based on their perception of themselves, their objectives, and their ability to raise questions and enact change when the world around them fails to resonate with their individual ethos [39,40,41,42,43]. The concept of political agency is both descriptive and prescriptive, encouraging individuals to seek emancipation and liberation from the unjust relationships that constrain them [40,44]. In the nursing domain, few studies use the concept of agency and there is almost a complete absence of studies theorising on the notion or exploring it empirically [42,45,46].

Organisational justice from the perspective of nurses and political agency in nursing are the two key concepts underpinning this study protocol. There are no previous studies in which the relationship between these two topics has been explored. However, there are recent studies in which, using a critical perspective, researchers focus on the actions of nurses facing conditions implicitly linked to organisational justice, such as exposure to continuous organisational changes [45] or recognition challenges to nurse practitioners’ role implementation and integration in clinical settings [46], among others. Organisational justice and political agency concepts allow us to highlight the need for nurses to be able to identify the forms of justice and injustice around them and to exercise their political agency to work towards greater justice in their workplaces. In order to advance our knowledge of these two themes, we must assess the current situation and propose alternatives to transform healthcare organisations into fairer environments for both the nurses working in them and the patients in their care [47].

### 1.1. Study Aim

The primary objective of this study is to explore nurses’ perceptions of organisational justice in public healthcare institutions in Majorca, Balearic Islands, Spain, and to analyse the ways in which they exercise their political agency to challenge the institutional order when it fails to reflect their professional ethos. In order to achieve this overarching objective, the study will (1) explore the ways in which nurses construct an identity with a view to understanding their professional ethos, (2) map elements of justice and injustice within public healthcare organisations to determine nurses’ approach to organisational justice, (3) describe the strategies of political agency used by nurses to remedy the forms of injustice identified, and (4) analyse how peripheral respondents (physicians and healthcare managers) perceive nurses’ experiences of organisational justice and injustice to understand this phenomenon in a deeper and comprehensive way.

### 1.2. Theoretical Framework

This study is situated within the critical social paradigm [48]. More specifically, it is informed by two critical conceptual frameworks: Nancy Fraser’s theory of justice and Hartmut Rosa’s theories of social acceleration and resonance.

In Nancy Fraser’s theory of justice, justice is understood as participatory parity or, in other words, as the economic, cultural, and political arrangements that ensure that all members of a community participate as peers (as equals) in social life [49]. This conceptualisation of justice will be helpful in interpreting the data for this study. Our understanding is that healthcare organisations are situated within a broader economic, cultural, and political system, operating within this system as social microsystems. As in wider society, the social status of the individuals who form part of these organisations is determined by forms of inequity that, in this case, hinder healthcare professionals’ equal access to resources to develop their practice, equal recognition of their role in organisations, and equal representation of their voices within them. This mediates social justice and injustice in healthcare organisations. Fraser’s theory of justice has been applied on social studies around issues of gender equity, welfare state, social protection, poverty, identity and exclusion to explain and respond to social injustices that ensnare women, LGTB people, and ethnic minorities [50]. In the area of healthcare research, only one paper has been published using Fraser’s theoretical approach [51]. In their study, Gibson, Lewando-Hund and Blaxter (2014) made use of the concept “strong and weak publics” from Fraser’s theoretical framework to explore parental participation in neonatal networks.

Meanwhile, Hartmut Rosa’s theory of social acceleration is based on the idea that a society or institution is modern when it operates in a mode of dynamic stabilisation, that is, when it systematically requires (material) growth, (technological) acceleration and (sociocultural) innovation to reproduce its structure and maintain the institutional status quo [50]. The primary consequence of social acceleration is alienation among individuals, who are unable to autonomously establish objectives, values, and paradigms by which to understand society, and practices to ensure that they live a good life [51]. For Rosa, alienation may be remedied through resonance or the construction of relationships with the world that are based on quality rather than quantity and possess the power to transform the lives of individuals and those around them [52,53]. Rosa’s theories of social acceleration and resonance have been applied to explore theoretical issues related to the logic of capitalist modernity, political democracy or ecological crisis [54], but also to more practical issues such as studies focusing on work-life balance [55]. These concepts will be useful for analysing the data generated in this study. Social acceleration is also present in healthcare organisations, affecting their functioning and the experiences, identities and practices of the workforce [56]. Resonance, on the other hand, points to the need for healthcare organisations to move beyond their current focus on productivity, performance and innovation in their working relationships to prioritise collective wellbeing and ethical care.

These two different theoretical approaches will be helpful to understand that justice within healthcare organisations depends on redistribution, recognition, representation and resonance conditions, and that nurses’ political agency will have a role to play in making these four conditions possible.

## 2. Methods

This is a qualitative ethnomethodological study using critical discourse analysis. Ethnomethodology is a branch of sociology whose primary aim is to understand the meanings assigned by individuals to their everyday lives and contexts by studying their narratives, words, and actions [57,58]. Critical discourse analysis, on the other hand, is a social research procedure that focuses on speech and text (language) to reveal the ways in which discourse constructs and mediates the naturalised sociocultural and political reality of a specific social and historical context [59].

### 2.1. Setting and Participants

The participants will be divided into two main groups: central respondents and peripheral respondents. On the one hand, the central respondents will be nurses working in primary care and in hospital settings run by public healthcare organisations on the island of Majorca (Spain), who will represent the main focus of the study. Five different nursing profiles have been selected for inclusion in the study: (1) currently employed clinical nurses; (2) currently employed middle-ranking nurses; (3) currently employed nurses in managerial or assistant managerial positions; (4) nurses in political and/or institutional roles; and (5) nurses in trade union roles. The selection of these five nursing profiles responds to the interest of including the perspectives of the different organisational positions in which nurses are represented in the context where this study will be carried out. On the other hand, the peripheral respondents will be professionals working alongside nurses and sharing most of their daily interactions with them, whose position in the organisation and active role in its decision-making allows them to exert significant influence over nurses’ perceptions of social and political justice and injustice in healthcare organisations. The reason for including peripheral respondents in this study responds to the fact that the issues of organisational justice and nurses’ political agency do not depend exclusively on nurses themselves—they are co-constructed issues. Thus, to understand these phenomena in a deeper and more comprehensive way, it may be useful to explore the sense that peripheral respondents give to nurses’ experiences. This second group includes two participant profiles: (1) currently employed physicians; and (2) managers of primary and specialist care organisations within the public healthcare system on the island of Majorca. The central and peripheral respondents must meet the following inclusion criteria: (a) be working at a public healthcare organisation at the time of data collection; (b) have at least 6 months’ experience of working at a public healthcare organisation; and (c) voluntarily agree to participate in the study and sign the informed consent form. No exclusion criteria for central and peripheral respondents are stated to ensure that the full diversity of individuals is included in this study [60].

### 2.2. Sampling and Recruitment

Participants will be selected using purposive sampling according to their narrative ability to illustrate the study phenomenon and snowball sampling will be used to generate additional subjects asking initial subjects to nominate peers [61,62]. Participants will be recruited via key informants at healthcare organisations and via a direct approach in the case of participants from groups that are underrepresented in the study context. A sample of 22–31 central respondents and 8–11 peripheral respondents is envisaged. The sample will be balanced by gender using gender distribution reports from the official associations to which the professionals working in the study setting belong. The final sample of participants will be dependent upon theoretical saturation of the data, so the initial sample size will be adjusted accordingly. Table 1 shows the initial sample of participants for inclusion in the study by profile and gender.

### 2.3. Data Generation Techniques

The data generation techniques that will be used in the study are semi-structured interviews, a sociodemographic questionnaire, and a field diary.

The semi-structured interviews will be carried out by the first author, a nurse researcher with prior experience conducting semi-structured interviews. Through open questions covering general and more specific topics, experiences of justice and injustice in healthcare organisations will be explored and the forms of agency exercised by the nurses participating in the study will be analysed [63]. The interview script will be designed by the authors, inspired by scripts proceeding from studies pursuing purposes of nurses’ emancipation [37,38] and in Fraser’s and Rosa’s theoretical framework to ensure theoretical coherence throughout the study. Table 2 shows an example of a preliminary script for the interview to be conducted with the central respondents. Minor adjustments will be made to this script to meet the particularities of each of the five nursing profiles. The interviews will be held in a location agreed with the participant, where their privacy and comfort are guaranteed and the conversation can take place without interruptions. All interviews will last for approximately one hour. They will be audio recorded with prior consent from participants and transcribed literally for subsequent analysis. Two potential participants (a female and a male employed clinical nurses) will be selected to pilot the interviews with a view to assessing the appropriateness of the questions, identifying possible issues relating to clarity and comprehension, ascertaining the emerging themes to confirm that they will allow the study objectives to be fulfilled, and identifying any personal difficulties involved in the process [64,65]. The two participants who will pilot the interviews will be included in the final sample. The interviews will be conducted in two phases. In the first phase, the interviews with the central respondents will be carried out in the order of the profiles listed above. In the second phase, the peripheral respondents will be interviewed. Interviews with core participants and peripheral participants will be different. The authors will design the script for the semi-structured interview for peripheral respondents once the results of the central participants’ interviews have been analysed and once the elements that need to be further explored from peripheral participants are identified.

A questionnaire will be used to gather sociodemographic information about the participants. This information will include data relating to gender, length of professional experience, current workplace, position within the organisation, and a telephone number or email address so that participants may be contacted if necessary. These data will be used as additional information to enhance contextual understanding of the study phenomenon.

The field diary will cover aspects relating to reflexivity and will include descriptive, methodological, and theoretical notes [66], which will be useful during the analytical process.

### 2.4. Data Analysis

A critical discourse analysis will be carried out using Fairclough’s approach. Fairclough proposes a three-dimensional model of critical discourse analysis built on the basis of the three levels from which discourse can be explored: the textual level, the level of discursive practice and the sociocultural level [59,67]. Crowe’s (2005) steps to carry out critical discourse analysis in nursing research will be useful in the analysis process of this study [68].

The semi-structured interviews will be analysed in a cross-cutting, continuous manner, which will achieve optimum efficiency once the data collection process is complete. The data analysis process will encompass the following steps: familiarisation with the data generated through repeated reading of the transcribed interviews to establish a series of pre-analytical intuitions or key overall ideas; identification, resulting in an index of general themes to facilitate data processing; indexing, which will consist of coding the data using inductive and deductive strategies; connection, grouping the codes generated in the previous phase into sub-categories and then into categories based on similar meaning; and interpretation, when the codes, sub-categories and categories will be analysed in light of the conceptual framework described above and the literature review that will accompany this study protocol [69,70,71]. These steps will first be applied to the transcriptions of the interviews with the central respondents and subsequently to those of the interviews with the peripheral respondents. We will create two different coding trees that will later be brought in dialogue to explore the convergences, divergences, and discursive tensions between them in an attempt to understand the phenomenon as a whole. Data will be codified independently by the three researchers. Once finished, all researchers will meet to compare their codification to identify convergences and divergences on coding and to agree to the final encoding. Field notes will be integrated into the interview transcripts and their content will be used in the analysis process [66]. ATLAS.ti 9 (Scientific Software Development GmbH, Berlin, Germany) software will be used if needed to organise and manage the data.

### 2.5. Rigour Assurance Procedures

Methodological rigour will be guaranteed through the use of validity strategies (e.g., member checking, literal and systematic transcription of interviews, the inclusion of negative or extreme cases, selection of quotes to illustrate results), reliability strategies (e.g., careful description of each phase in the study, including the context, the final sample of participants, the sampling strategy, recruitment, data generation, data analysis), and strategies for internal and external transferability (e.g., detailed description of participants’ characteristics and study context to allow the study to be reproduced with similar participants and contexts; use of broad inclusion criteria to obtain a final sample of participants that encompasses the greatest possible number of voices, perspectives, and actions; data analysis until saturation is reached, and use of a solid conceptual framework based on normative concepts). In addition, the respondents, researchers, and theories will be triangulated, thus revealing different dimensions of a research phenomenon and ensuring a rich and robust data analysis, and the researchers will engage in reflexivity throughout the entire research process [72,73].

### 2.6. Ethical and Regulatory Considerations

This study has been approved by the Research Ethics Committee of the Balearic Islands (IB 4013/19 PI) and will be conducted in accordance with the principles of the Declaration of Helsinki. Future participants will receive verbal and written information about the study and will sign an informed consent form before the interview process begins. Participation will be voluntary. Data for all participants will be anonymised and both personal data and interview data will be treated as confidential, stored securely, and accessed only by the research team. Participants may withdraw their informed consent at any time without penalty. Participation in the study will neither generate costs nor be rewarded with financial compensation.

## 3. Discussion

This study protocol describes a targeted research project whose primary objective is to explore nurses’ perceptions of organisational justice in public healthcare institutions in Majorca, Balearic Islands, Spain, and to analyse the ways in which they exercise their political agency to challenge the institutional order when it fails to reflect their professional ethos. Focusing on nurses will allow us to understand their perspectives regarding their status in healthcare organisations, the elements explaining this status (relating to resource distribution, professional recognition, and representation in decision-making), their aspirations when it comes to building fairer healthcare organisations, and their ideas and/or concrete actions taken to make this a reality, allowing them to look after their own interests and improve the care they offer their patients.

Regarding the first specific objective, we expect that exploring nurses’ identity construction will help us to understand how their ways of being and acting influence their greater vulnerability to various forms of organisational injustice. Regarding the second specific objective, we expect that mapping justice and injustice elements in healthcare organisations will help us to understand how nurses are positioned in a less privileged status in comparison with other healthcare professionals, and on what elements more efforts to build fairer and resonant healthcare organisations should be invested. Regarding the third specific objective, we expect that exploring nurses’ political agency strategies will help us to understand how they perceive their possibilities of emancipation in healthcare organisations. Finally, regarding the fourth objective, we expect that exploring peripheral respondents’ narratives will help us to better understand nurses’ narratives, but also to identify building fairer and resonant healthcare organisations main challenges.

### 3.1. Strengths

This project has three main strengths. Thematically speaking, political agency is an emerging theme in the nursing domain, while organisational justice from a macrostructural perspective (related to the broader economic, political, social, and cultural context) is an emerging theme in the field of healthcare services. Research into these themes is increasingly necessary and this study aims to fill a gap in the existing scientific literature. Methodologically, the inclusion of peripheral respondents in the study to obtain a deeper, more global understanding of the narratives presented by the central respondents is innovative in the field of qualitative research techniques. Finally, theoretically speaking, the data generated as part of this study will be analysed in line with the philosophical approaches set out by Nancy Fraser, whose work has only been used in one other article in the field of healthcare [51], and Hartmut Rosa, whose work has not yet been used in this field.

### 3.2. Limitations

The limitations of the study protocol presented here relate to some of its methodological considerations. Eligibility criteria for participants do not include exclusion criteria related to age, years of experience, marital status, or contractual conditions. Future studies could incorporate these determinants as study variables, thus advocating for intersectionality in research. The sole use of interviews as the data collection technique will allow the participants’ discourses to be analysed but not their actual practices. In the future, a complementary study supplementing conversational techniques with naturalist techniques (observation) could be used to conduct an ethnographic study on this subject. Snowball sampling is defended by some scholars as a sampling technique affecting the representativeness of the sample of participants [61]. Participants will not be selected randomly but will be dependent on the references made by previous respondents. Interviewees with smaller networks may be unrepresented. Previous respondents can recommend other potential participants relatively similar to themselves, based on criteria of friendship or proximity in the social network. Snowball samples may be more biased towards those more cooperative participants who want to participate in the study, a common limitation also in random sampling [60,61,62].

### 3.3. Dissemination Plan

The research team intends to disseminate the findings of the study using four main strategies: publication of articles in peer-reviewed scientific journals, participation in national and international conferences, organisation of information sessions aimed at nurses and decision-makers, and production of audiovisual material to raise awareness of the project.

## 4. Conclusions

The primary objective of this study is to explore nurses’ perceptions of organisational justice in public healthcare institutions in Majorca, Balearic Islands, Spain, and to analyse the ways in which they exercise their political agency to challenge the institutional order when it fails to reflect their professional ethos.

The results of the study are expected to provide new knowledge with interdisciplinary repercussions for clinical practice, politics, research, and education. In the clinical practice sphere, our findings will help nurses to identify the conditions to which they are entitled as professionals in order to perform their duties to a high standard and on an equal footing with other professionals. In the political sphere, they will cast light on ways of implementing strategies to ensure that nurses encounter the necessary conditions for resource distribution, professional recognition and representation to build fairer working environments that are safer both for those who work in them and for those receiving care. In the research sphere, they will open up new avenues for research, as political agency in nursing has seldom been explored, organisational justice is of the utmost importance in allowing the challenges emerging around the world to be dealt with effectively, and the conceptual frameworks selected to interpret the data are highly innovative in the healthcare field. Ethnographic research using participant observation as a data collection method could help us to continue understanding how nurses enact their political agency in the natural context, and participatory action research could contribute to implement changes pointed out by nurses to transform healthcare organisations into fairer and resonant settings. Finally, in the educational sphere, our findings will enable the development of educational strategies that foment a strong desire among future nurses to advocate for their own interests and those of their patients.

## Figures and Tables

**Table 1 ijerph-18-09110-t001:** The initial sample of participants for inclusion in the study by profile and gender.

Respondents	Profiles	Female	Male	Total
Central respondents	(1) Currently employed clinical nurses	10–12	2–4	12–16
	(2) Currently employed middle-ranking nurses	3–4	1–2	4–6
	(3) Nurses in managerial or assistant managerial positions	1–2	1	2–3
	(4) Nurses in political and/or institutional roles	1–2	1	2–3
	(5) Nurses in trade union roles	1–2	1	2–3
Peripheral respondents	(1) Currently employed physicians	3–4	3–4	6–8
	(2) Managers of healthcare organisations	-	2–3	2–3

**Table 2 ijerph-18-09110-t002:** The script for the semi-structured interview for clinical nurses.

What does being a nurse mean to you?
Is your way of nursing different to the way you would like to nurse? In what way?
What is the first thing that comes to mind if I ask you about the problems you are currently experiencing as a nurse? What about the problems faced by the profession as a whole?
If you had to give me one noun, adjective or expression to define the experiences and situations we’ve discussed, what would it be?
As a nurse, do you believe you have the power, influence, and ability to shape the current situation and develop strategies to change or transform your current working conditions? How?

## Data Availability

The datasets generated as part of the study will not be made available to the public for ethical reasons and to protect participants’ identities.

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
