# Peer review of "Organisational Justice and Political Agency among Nurses in Public Healthcare Organisations: A Qualitative Study Protocol"

_ijerph, 2021, doi:10.3390/ijerph18179110_

Round 1

Reviewer 1 Report

This is an interesting proposal by the authors whose study seeks to examine explore Majorca nurses’ perceptions of organizational justice in public healthcare institutions and analyze how they exercise their political agency to challenge the institutional order when it fails to reflect their professional ethos. However, some further improvements are possible to make the study protocol even better and allow for more effective research work.

  1. This is one of the first studies to address organizational justice in healthcare organizations from a macrostructural  perspective and to explore nurses’ political agency

The authors could consider identifying those previous research and indicating the gaps to be fulfilled by the proposed research.

  1. Five different nursing profiles have been selected for inclusion in the study

The authors could further explain the rationale for the five profiles.

  1. In the discussion and conclusion sections, further challenges could be explored.

Reviewer 2 Report

The paper focuses on an interesting topic and is presented as one of the first studies to address organisational justice in healthcare organisations from a macrostructural perspective and to explore nurses’ political agency.

The study protocol describes an innovative methodology that provides for the inclusion of peripheral respondents to obtain a global understanding of the organisational justice and political agency among nurses. Moreover, data analysis follows the philosophical approaches set out by Nancy Fraser and Hartmut Rosa, until now little used in the field of healthcare.

Although an in-depth author’s knowledge of this new topic is evident, I have to point out some improvements concerning the representativeness of the sample and the semi-structured interview form, that need to be considered before publication.

A point-by-point analysis is provided below:

INTRODUCTION:

  • Row 91 and 121: the use of the term “(in)justice” is not clear, please specify if it is taken from other authors and its meaning;

METHODS

  • Row 170-171: a total of 22-31 central respondents and 8-11 peripheral respondents seems to be insufficient to guarantee the sample representativeness, also in a qualitative research. Table 1 shows many sample groups consisting of only 1 or 1-2 individuals (i.e., male middle-ranking, female manager nurses). I recommend reviewing the number of respondents to ensure adequate representativeness of all groups.
  • Row 181: please specify in the sentence the “first author” role/qualification and whether she has experience in conducting semi-structured interviews;
  • Row 184-187: the authors state that “the script for the semi-structured interview will be modified to reflect the profile of the interviewee and the data from the interviews with other clinical nurses.”. This could be an important flaw because it introduces variability and subjectivity of asking questions and giving answers. Please specify if this method has already been adopted by other authors and provide further details.
  • Row 191: please specify the profile of the two potential participants who will pilot the interviews and confirm that they are nurses (male or female? Employed or managers?). Will they be excluded from the sample?
  • Table 2: to ensure reproducibility, please report whether the questions were taken (or inspired) from another work or if they were completely designed by the authors and provide further details.

Reviewer 3 Report

First of all, I would like to congratulate the authors for the study they have designed and for the clarity of their presentation. However, I would like to add a few comments in the hope that the authors can improve their study.

Theoretical framework. It is well synthesized and summarized. I miss some results of the empirical evidence they point out that goes in line with what they want to demonstrate, although the studies they refer to do not focus on the population on which the authors are going to focus.

Objectives. The general objective is well stated. However, for a better reading of the paper, it would be convenient to break down the specific contents (1, 2, 3, etc.). In any case, a) the objective(s) related to the sample of peripheral participants should be included and b) the specific contents should be reflected upon in greater depth. They start from two different theoretical approaches and use two different samples (central and peripheral participants). Therefore, I believe that the specific objectives should also have a distinctive character.

For a better reading, break down point 2.1. Setting and participants in a subsection and sampling and recruiment in another subsection.

Subsection Setting and participants. I have some doubts about the inclusion/exclusion criteria. A) Is there no exclusion criterion? If so, please specify it and why; B) Is age or whether nurses work in private institutions or their marital status or whether their contract is permanent/discontinuous not considered as inclusion criteria? I believe that not considering these aspects could affect the criteria of homogeneity and hetereogeneity of the sample, in addition to biasing and hindering the generalizability of future results. In any case, all these points should be explained.

Subsection sampling and recruitment. Specify the biases of a snowball sample and whether the three techniques used for the sample correspond to the technique used in the qualitative methodology, i.e., a convenience sample.

Data Generation Techniques: Missing quotes from line 184 to line 198. There is a large amount of literature on how data collection should be done in semi-structured interviews. The next question I have in this section is whether authors are going to conduct the same interview with core participants and peripheral participants. I think they should be different. First of all because they are based on two different theoretical approaches, one seems to be more micro and correspond to central participants and the other more macro which seems to relate to peripheral participants. If so, please explain and add the questions that would correspond to peripheral participants.

2.3. Data Analysis. This is the weakest section of the research. There is a huge amount of theoretical and empirical evidence on this subject. There is nothing specified about Grounded Theory or the constant comparison method, basic in qualitative analysis. This topic should be explored in more depth and not be based only on the article by Pope, Ziebland and Mays (2000), which is a brief synthesis of the steps to be followed in qualitative analysis. In addition, it is necessary to specify: a) whether or not software will be used to analyze the analyses, b) how inter-rater reliability will be found, c) specify the advantages of using triangulation between different sources for the results of the study, d) how the field notes will be integrated with the study data and e) add a figure that represents the analysis to be carried out on two axes.

Discussion. It only reflects generalities. It would be important for this section to show conclusions based on the specific objectives. Perhaps it would help to include a previous section (e.g., expected results) in which they specify what they expect to obtain.

Round 2

Reviewer 2 Report

My suggestions have been fully considered and sufficient improvements were made by the authors.

To ensure reproducibility of the results in further studies, I still have some doubts about the representativeness of the sample (in particular the lower number of participants of certain profiles). This could be a limit of the study.

Nevertheless, in my opinion the manuscript is now eligible for publication in IJERPH.

Congratulations and good luck!

Reviewer 3 Report

Dear Authors,
Thank you very much for considering the proposed modifications. And, of course, congratulations for the depth and thoughtfulness of your study protocol.

Best regards